# Spatio-Temporal Analysis of Cultivated Land from 2010 to 2020 in Long'an County, Karst Region, China

**Jianhui Dong** [1], **Wenju Yun** [2,*], **Kening Wu** [1,3], **Shaoshuai Li** [2], **Bingrui Liu** [1] and **Qiaoyuan Lu** [4]

1. School of Land Science and Technology, China University of Geosciences, Beijing 100083, China
2. Center for Land Reclamation, Ministry of Natural Resources, Beijing 100035, China
3. Key Laboratory of Land Consolidation, Ministry of Natural Resources, Beijing 100035, China
4. Guangxi Xin Hang Land Planning and Consulting Co., Ltd., Nanning 530025, China
* Correspondence: yunwenju@lcrc.org.cn

**Abstract:** Spatio-temporal changes in cultivated land have a profound impact on food security and sustainable development. However, existing studies on spatio-temporal changes in cultivated land mostly focus on single factors, for instance quantity, quality and ecology, that cannot comprehensively reflect the changes in total production capacity and the sustainability of cultivated land. This study aims to construct a comprehensive analysis approach and to provide a reference basis for a comprehensive analysis of the extent of changes in overall cultivated land food-production capacity and the formulation of cultivated land conservation-related policies. This comprehensive analysis method constructed from three dimensions: quantity, production capacity and ecology, fully reflects the changes in the total amount, structure, rate of change, spatial distribution, quality, total production capacity and sustainability of cultivated land. The results from the application of this approach to Long'an County, Guangxi Province, China demonstrate that: (a) from 2010 to 2020, the total amount of cultivated land in Long'an County decreased sharply by 30.83%, accounted for mainly by the conversion into orchards, forest land and other garden land; (b) the quality of cultivated land improved by 2.71% on average, mostly in relation to natural factors; (c) the total food-production capacity of cultivated land decreased by 28.96% on average, mainly due to the decrease in the area of cultivated land; (d) both the ecological grade and the sustainability of cultivated land decreased slightly; (e) the barycenter of cultivated land migrated 3.3 km to the ecologically sensitive areas in the west, and the patch size of cultivated land decreased from an average of 2.60 hectares/pc in 2010 to that of 1.34 hectares/pc in 2020, suggesting increased fragmentation of cultivated land; and (f) the patch regularity of cultivated land decreased from 2.08 in 2010 to 1.76 in 2020, showing improved patch regularity and slightly better adaptability to mechanization. There were two main reasons for the lower, total food production capacity in Long'an County: first, the low comparative income of grain cultivation, because of which farmers spontaneously adjusted the agricultural cultivation structure to pursue high returns; and second, the lack of targeted government policies to protect cultivated land. In general, this comprehensive analysis method is applicable to other provinces in China or other regions abroad to provide a reference basis for a comprehensive understanding of changes in the food production capacity of cultivated land and the formulation of policies on cultivated land protection.

**Keywords:** land-use change; cultivated land quality; production capacity of cultivated land; ecology of cultivated land; fragmentation of cultivated land; Karst region

## 1. Introduction

A thorough understanding of the spatial characteristics of cultivated land changes on a global scale [1,2] is required to ensure food security. The COVID-19 pandemic unleashed a global food security crisis and nearly one in three people in the world (2.37 billion) did not have access to adequate food in 2020—an increase of almost 320 million people in just one

year [3]. Economic development, population growth and accelerated urbanization [4–7] have led to a dramatic expansion of construction land [7,8] and the loss of large amounts of high-quality cultivated land [6,9]. A decrease in cultivated land area, a decline in cultivated land quality, water scarcity and climate change [10–14] bring great challenges to food security and sustainable development. China's per capita cultivated land is less than 40% of the world average, its freshwater resources account for only 9% of the world average and the superior cultivated land area amounts to less than 30% of the total cultivated land. While China's urbanization rate increased by 1%, the cultivated land area decreased by about 2 million mu [15]. The decline in cultivated land poses a serious challenge to China and directly threatens its food and national defense security. However, the momentum of the reduction in cultivated land remains unabated in China, and according to the data of the Third National Land Survey published in 2021, the national cultivated land has decreased by 113 million mu in ten years since 2009, mainly due to agricultural restructuring and afforestation where 112 million mu of cultivated land became forest land and 63 million mu became garden land [16]. Therefore, considering the multiple factors such as unprecedented changes in a century, intensified international competition, political turmoil, climate change and the development needs of the domestic economy [17], it is of great practical significance to scientifically understand the laws of changes in the spatio-temporal patterns of cultivated land and to reasonably coordinate the relationship between economic development and cultivated land protection.

The study of spatio-temporal changes in cultivated land has been a hot issue in the field of cultivated land conservation and food security, and scholars at home and abroad have conducted a lot of studies on the changes in cultivated land patterns in different periods [18–23]. In terms of research scale, the existing studies focus on the global scale [2,24–27], national scale [28–30], provincial scale [8,31–33] and municipal and county scale [34–36]. In terms of research content, most of the existing studies focus on one aspect of change [37], such as in the quantity of cultivated land [27,30,35,38,39], the quality of cultivated land [34,40,41], the ecology of cultivated land and the level of pollution, whereas there are few studies covering a comprehensive analysis of cultivated land change. In terms of research methods, few studies have relied on the trinity approach to comprehensively analyze spatial and temporal changes in cultivated land. The main methods for studying the amount of cultivated land are the Remote Sensing Image Interpretation Method, the Land Use Dynamicity model, the Land Use Conversion Matrix, the Standard Deviation Ellipse, the Spatial Correlation Analysis method, etc. As far as the research method of cultivated land quality is concerned, the comprehensive index method is mainly used to construct the evaluation index system, determine the index scores and weights and calculate the index scores. The main research methods for cultivated land ecology are the InVEST model, the landscape pattern index and Landscape Pattern Metrics. Zhou et al. (2021) studied the spatio-temporal land-use change and its driving forces on the Loess Plateau in northern Shaanxi from 1980 to 2020 and found that construction land, forest land and grassland continued to increase in size, but the cultivated land decreased significantly [20]. Yuan et al. (2022) studied the effects of urban sprawl on cultivated land loss and fragmentation in major grain-producing regions of China and found that urban sprawl was the main cause of cultivated land loss and fragmentation [19]. The previous studies have focused on the following areas: (1) studies on the change in the amount of cultivated land in different periods showed that construction occupancy leads to the loss of large amounts of high-quality cultivated land [7,10]. In addition, studies on different regions found that cultivated land loss is different in rate and significance among regions; (2) construction occupies large amounts of cultivated land and causes fragmentation [19] and an overall decline in the quality of cultivated land [21]; (3) studies of ecological changes in cultivated land mainly focused on the landscape pattern of cultivated land [38]. Most of the new cultivated land is far from settlements and close to ecologically sensitive areas, leading to increased farming costs and threats to sustainable land-use. This research mainly explores the changes in the spatial pattern, the quality and ecology of cultivated land in different

periods through the land use dynamicity model, land use conversion matrix, the integrated index method and spatial autocorrelation analysis.

At this stage, research on spatio-temporal changes of cultivated land have the following shortcomings: first, most existing studies have focused on economically developed regions in eastern China [32,42], while fewer studies have considered this issue from a county-scale perspective in karst [43] and underdeveloped regions [44–46] in southwest China. In China, rapid economic development has led to an increasingly prominent conflict between people and land, and much more prominent resource constraints and cultivated land loss in underdeveloped county-level regions. Less developed county areas are the major concentrated producers of agricultural products, an important part of the food security plan, and an important arena of rural revitalization in China; therefore, it is of great practical significance to conduct research in underdeveloped county-level areas. Second, existing studies are carried out using the data mainly derived from remote sensing monitoring and interpretation [47], while few are completed using high-precision National Land Survey data. The comparison between the commonly used land use and land cover remote-sensing monitoring data with a resolution of 30 m × 30 m [25,48–50] turns out that this level of accuracy could reflect neither the changes in cultivated land that are small in area nor the precise conversion from cultivated land into other types of land. Finally, we chose the much more accurate National Land Survey data. In this context, the use of high-precision Land Survey data to address the spatio-temporal changes of cultivated land at the county level in Karst and underdeveloped areas of southwest China, and then to explore whether regional cultivated land resources will continue to play a role in ensuring local economic development, increasing farmers' income and supplying agricultural products, is of great significance for achieving high-quality regional development.

The study area, a karst region in the southwest of the country, is located in Long'an County, Guangxi Province, China. The data of the third Land Survey shows that the area of cultivated land in the mountain or plateau areas of southwest China has decreased more from 2010 to 2020. Long'an, a southwest county of a southwest province in China, has seen its cultivated land reduce extensively during the ten years. The analysis of the reasons for the decrease in cultivated land in Long'an County and the extent of its change in food production capacity during the 10-year period is important for understanding the change in the food production capacity of cultivated land and the formulation of policies related to cultivated land-protection in China. This paper relied on the land use dynamicity model and land use conversion matrix to illustrate the extent and structural characteristics of cultivated land-use change, the standard deviation ellipse and landscape metric to characterize the structural change and fragmentation of cultivated land, the integrated index method to evaluate the quality change of cultivated land and the spatial autocorrelation analysis to illustrate the aggregation of cultivated land production-capacity. By applying the approach to the study area and analyzing the spatio-temporal changes to cultivated land in Long'an County that occurred during the past ten years, we discovered the spatio-temporal patterns of cultivated land changes (Figure 1).

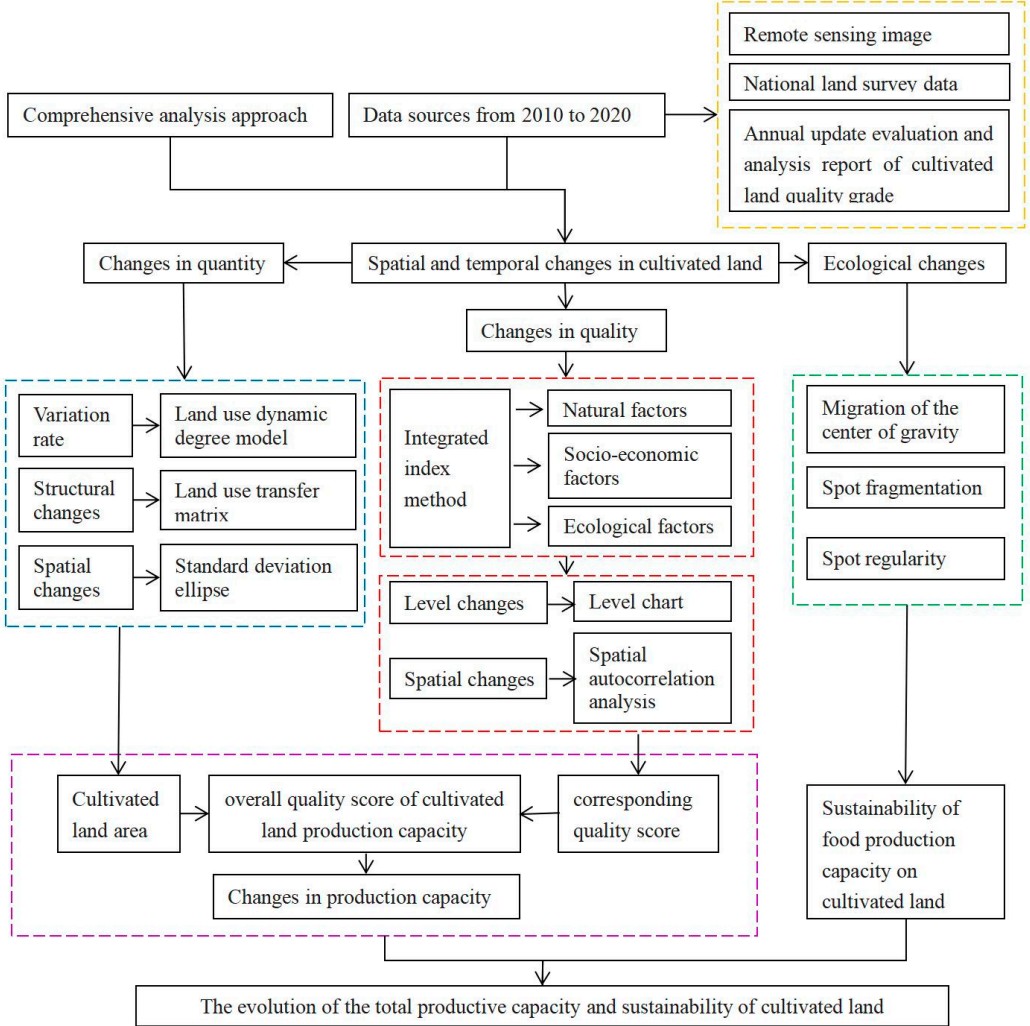

**Figure 1.** The research frame diagram.

## 2. Data and Methods

### 2.1. Study Area

The study area, Long'an County, is located in the karst region of southwest part of Guangxi Province of China (Figure 2). The county, running for a maximum horizontal distance of 77 km between its east and west and a maximum vertical distance of 56 km between its north and south and covering a total area of 2307.26 km$^2$, is sited in the south of the Tropic of Cancer and therefore under the humid southern subtropical monsoon climate where it has an average annual rainfall of 1301 mm, an annual average temperature of 21.8 °C, a monthly average maximum temperature of 28.4 °C and a monthly minimum average temperature of 13.2 °C. Long'an County is located in the karst mountains of southwest Guangxi, surrounded by high mountains on both sides and lying low in the middle along the valley of the Youjiang River. It reposes on the plain in the shape of a belt of isolated peaks and monadnocks with a north–west to south–east orientation and on peak cluster depressions and valleys in the southwest of Dujie Township, Buquan Township, and Ping Shan Township. The entire terrain slopes slightly towards the southeast, composed of low and medium mountains and low hills of craggy rocks in the northeast and a plain of valleys and monadnocks in the central part and coursed through in the middle of the county by the Youjiang River from the northwest to the southeast. Hilly land, karst land, plain, middle and low mountains and water account for 48.29%, 31.5%, 12.44%, 1.6% and 6.11%, respectively, of the county. Therefore, it is a typical mountainous county. It consists of 4 townships and 6 towns, 118 administrative villages and 14 communities. As of the end

of 2021, Long'an County had a household registered population of 420,500 from 13 ethnic groups including Zhuang, Han, Miao, Yao and others. The small-scale research of Long'an county, a karst region in southwest China and an important agricultural base in Guangxi Province, is reflective of the regional spatio-temporal variation patterns of cultivated land, and important for cultivated land conservation and socio-economic development.

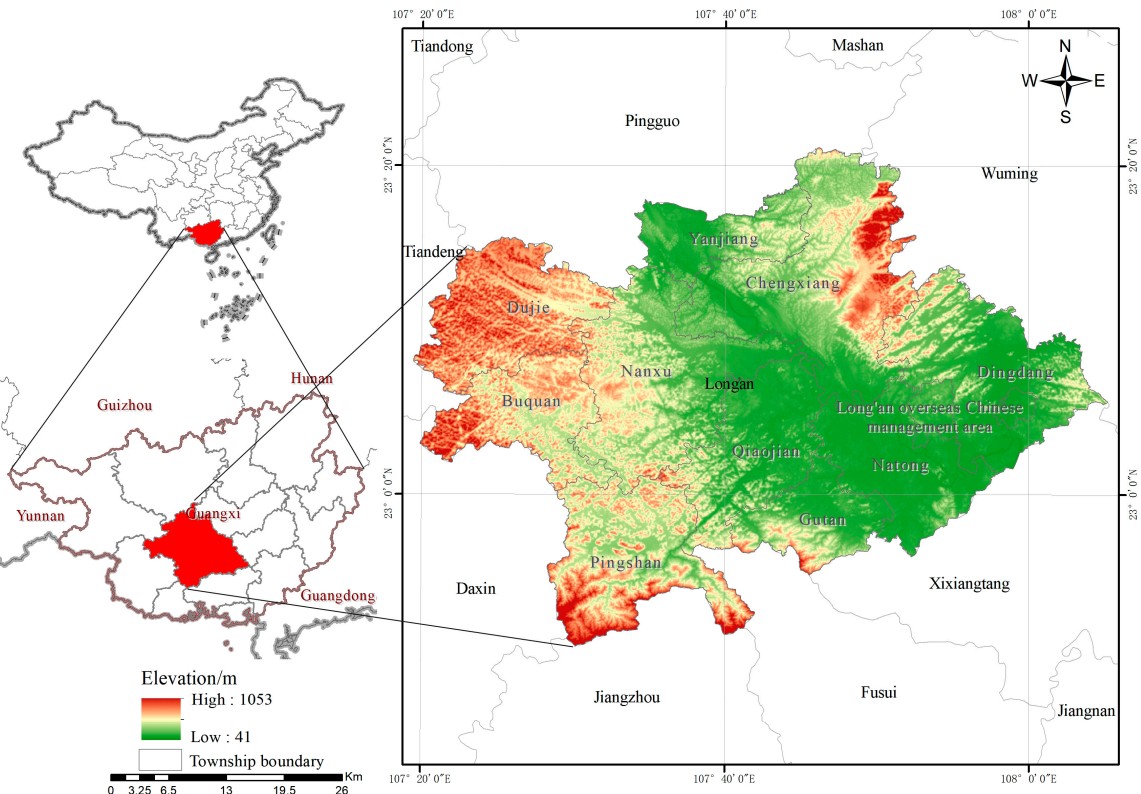

**Figure 2.** Location of the study area.

### 2.2. Data Sources

The land use data used in this paper were from the National Land Use Survey between 2010 and 2020. Soil data for cultivated land quality evaluation were derived from the analysis report in the 2013 and 2020 annual update and evaluation of cultivated land quality grade in Long'an county. Slope data were extracted from the 30 m resolution DEM data downloaded from the geospatial data cloud platform (http://www.gscloud.cn/), accessed on 28 October 2022.

### 2.3. Methods

#### 2.3.1. Analysis of Changes in the Amount of Cultivated Land

(1) Land use Dynamicity Model

The land use dynamicity model describes the quantitative changes of a certain land-use type in a certain period [27,31,32,44,51]. Positive (negative) values indicate the net increase (net decrease) in land area. The absolute value of land use dynamics indicates the extent of change in a particular land-use type. It is clear that the higher the absolute value is, the higher the dynamicity is, the more drastic and less stable the change in the particular land-use type is. The calculation formula is:

$$K = \frac{U_b - U_a}{U_a} \times \frac{1}{T} \times 100\% \tag{1}$$

where $K$ is the dynamicity of the land-use type during the study period; $U_a$ and $U_b$ denote the number of land-use types at the beginning and end of the study, respectively. $T$ is the length of the study period, and when $T$ is set to a certain number of year(s), the value of $K$ is the annual change rate of a certain land-use type in the study area.

(2)　Land use Conversion Matrix

The land use conversion matrix reflects the degree and direction of transitions between structural features and land-use types [19,32,42,43], thus reveals the spatio-temporal evolution of land-use patterns. Its formula can be expressed as follows:

$$S_{ij} = \begin{bmatrix} S_{11} & \cdots & S_{1n} \\ \vdots & \ddots & \vdots \\ S_{n1} & \cdots & S_{nn} \end{bmatrix} \tag{2}$$

where $S_{ij}$ is the area changed from land-use type $i$ into land-use type $j$; $n$ is the number of land-use types; $i$ and $j$ represent the land-use type before and after the conversion, respectively.

(3)　Standard Deviation Ellipse (SDE)

SDE accurately reflects the spatial distribution pattern and evolutionary characteristics of the study object [19,52,53], thus reveals the aggregation state and the offset trend of cultivated land. Its main parameters include the barycenter, the azimuth and the long and short axes of the ellipse, where the elliptical barycenter characterizes the barycenter of geographic elements and their migration directions, distances and velocity trajectories, the azimuth angle represents the main trend of their distribution on the two-dimensional space, the long half-axis of the ellipse indicates the direction of cultivated land distribution, and the short half-axis, the extent of cultivated land distribution. In this paper, an SDE is constructed to reflect the change in the spatio-temporal patterns of cultivated land. These three parameters are calculated as follows:

$$\overline{X} = \sum_{i=1}^{n} w_i x_i / \sum_{i=1}^{n} w_i, \ \overline{Y} = \sum_{i=1}^{n} w_i y_i / \sum_{i=1}^{n} w_i \tag{3}$$

$$\tan \alpha = \frac{\left[ \left( \sum_{i=1}^{n} w_i^2 x'^2 - \sum_{i=1}^{n} w_i^2 y'^2 \right) + \sqrt{\left( \sum_{i=1}^{n} w_i^2 x'^2 - \sum_{i=1}^{n} w_i^2 y'^2 \right) + 4 \sum_{i=1}^{n} w_i^2 x'^2 y'^2} \right]}{2 \sum_{i=1}^{n} w_i^2 x' y'} \tag{4}$$

$$\delta_x = \sqrt{\left( \sum_{i=1}^{n} w_i x' \cos \alpha - w_i y' \sin \alpha \right)^2 / \sum_{i=1}^{n} w_i^2} \tag{5}$$

$$\delta_y = \sqrt{\left( \sum_{i=1}^{n} w_i x' \sin \alpha - w_i y' \cos \alpha \right)^2 / \sum_{i=1}^{n} w_i^2} \tag{6}$$

where $\overline{X}$ and $\overline{Y}$ are the horizontal and vertical coordinates of the standard deviation of the ellipse, respectively; $(x_i, y_i)$ is the coordinates of feature $i$; $w_i$ is its corresponding weight; $\alpha$ is the orientation angle; $\delta_x$ and $\delta_y$ are the standard deviations along the $x$ and $y$ axes, respectively; $x'$ and $y'$ denote the difference between the coordinates of the barycenter and the coordinates of the geographic element.

### 2.3.2. Analysis of Changes in Cultivated Land Quality

(1)　Evaluation index system

In this study, the cultivated land patches in two periods from the National Land Survey were selected as evaluation units. The index system for evaluating cultivated land quality was developed by the principles of comprehensive analysis, dominant factors, income differences and spatial variability and based on data availability. Thus, based on the theories of agricultural location theory, land rent theory and ecological service value, as well as the actual situation of Long'an County, and by reference to the *Gradation Regulations*

for Agricultural Land Quality (GB/T 28407-2012), *Regulations for Gradation on Agricultural Land* (GB/T 28405-2012) and *Cultivated Land Quality Grade* (GB/T 33469-2016), evaluation indicators were selected at the three levels [54] of natural factors, socioeconomic factors and ecological and environmental factors [38] in the six aspects of topography and landscape, soil conditions, infrastructure conditions, farming conveniences, ecological quality and environmental conditions [55,56]. Correlation analysis was performed among the indicators to eliminate those with a strong correlation, and finally, a total of 14 indicators were selected to evaluate the quality of cultivated land, including average slope, effective soil layer thickness, surface soil texture, organic matter content, soil PH, irrigation guarantee rate, drainage conditions, field road density, cultivation distance, patch size, patch regularity, ecological land coverage, soil erodibility K and soil pollution level (Table 1) [57–59].

**Table 1.** Indicator system for evaluating cultivated land quality in Long'an County.

| Decision-Making Level | Target Level | Criterion Level | Indicator Level | Selection Basis | Effect |
|---|---|---|---|---|---|
| Cultivated land Quality | Natural Factors | Terrain topography | Average slope | Field flatness | − |
| | | Soil conditions | Effective soil layer thickness<br>Surface soil texture<br>Organic matter content<br>Soil PH | Crop root growth<br>Soil permeability<br>Soil fertility<br>Soil acidity and alkalinity | +<br>+<br>+ |
| | Socio-economic factors | Infrastructure conditions | Irrigation guarantee rate<br>Drainage conditions<br>Density of field roads | Drought resistance<br>Drainage capacity<br>Field accessibility | +<br>+<br>+ |
| | | Farming Conveniences | Cultivation distance<br>Patch size<br>Patch regularity ($D_{11}$) | Ease of management<br>Mechanizable degree<br>Arable potential | −<br>+<br>− |
| | Ecological factors | Ecological quality | Ecological land coverage<br>Soil erodibility K value | Ecological adaptive capacity<br>Soil and water conservation capacity | +<br>− |
| | | Environmental conditions | Soil pollution level | Soil heavy metal pollution | − |

Data acquisition: average slope was obtained by DEM calculation; effective soil thickness, surface soil texture, organic matter content, soil PH value, irrigation guarantee rate were obtained by the updates of quality grades of Long'an County's cultivated land; drainage conditions were obtained by overlaying the cultivated land layer with the ditch layer and then calculating the ditch length, and field road density, by overlaying the cultivated land layer with the field road layer and then calculating the lengths of the roads and ditches. Cultivation distance was obtained by proximity analysis; patch size was obtained by using the area field of the land type layer; patch regularity was obtained by using the patch shape index; ecological land coverage was calculated by selecting three main ecological land types, namely forest land, grassland and water area, from the land type map per village, and then obtaining the attribute values per village; soil erodibility K value was obtained by inputting the eight heavy soil types, namely As, Hg, Cu, Zn, Pb, Cd [60] and Cr, extracted from the national soil type data (1:1,000,000) into the EPIC model; soil pollution degree was calculated with the Nemero integrated index based on the eight heavy metals selected, namely As, Hg, Cu, Zn, Ni, Pb, Cd and Cr.

Evaluation index assignment: average slope, effective soil layer thickness, surface soil texture, organic matter content, soil PH and irrigation guarantee rate, were quantified with reference to the index grading assignment rules under the technical specifications for updating the quality of cultivated land in Guangxi (Table 2); drainage conditions, density of field road, cultivation distance, patch size, patch regularity, ecological land cover, soil erodibility K value, and soil pollution level, were determined with reference to the *Gradation Regulations for Agricultural Land*. The criteria and weights of the quality grades of cultivated land in this study area [21,55,57,61] are shown in Table 2.

**Table 2.** Criteria and weights of quality evaluation indicators for cultivated land in Long'an County.

| Evaluation Indicators | Score | | | | | Weights |
|---|---|---|---|---|---|---|
| | 100 | 80 | 60 | 40 | 20 | |
| Average slope | ≤2 | (2, 5] | (5, 8] | (8, 15] | >15 | 0.16 |
| Effective soil layer thickness | ≥100 | [70, 100) | [50, 70) | [25, 50] | <25 | 0.07 |
| Surface soil texture | Loam | Sandy loam | Clay | Sandy soil | Gravelly soil | 0.09 |
| Organic matter content | ≥40 | [20, 40) | [10, 20) | [6, 10) | <6 | 0.12 |
| Soil PH | [6.5, 7.5) | (4.5, 6.5) | [7.5, 8.5) | ≤4.5 | ≥8.5 | 0.05 |
| Irrigation guarantee rate | 1 | 2 | 3 | 0 | 4 | 0.11 |
| Drainage conditions | 1 | 2 | 3 | 0 | 4 | 0.04 |
| Density of field roads | [0.85, 1) | [0.60, 0.85) | [0.45, 0.65) | [0.20, 0.45) | <0.2 | 0.05 |
| Cultivation distance | [0, 0.2] | (0.2, 0.5] | (0.5, 1] | (1, 1.8] | >1.8 | 0.06 |
| Patch size | ≥100 | [50, 100) | [10, 50) | [5, 10) | <5 | 0.03 |
| Patch regularity | ≤1.25 | (1.25, 1.8] | (1.8, 2.5] | (2.5, 3.7] | >3.7 | 0.02 |
| Ecological land coverage | ≥0.7 | [0.5, 0.7) | [0.3, 0.5) | [0.17, 0.3) | <0.17 | 0.04 |
| Soil erodibility K value | ≤0.0024 | (0.0024, 0.0026] | (0.0026, 0.0028] | (0.0028, 0.003] | >0.003 | 0.06 |
| Soil pollution level | 0 | Mild | Moderate | 0 | Severe | 0.10 |

(2)  Determination of evaluation index weights and establishment of evaluation model

The evaluation index weights were determined using principal component analysis and hierarchical analysis [61,62] in combination with both quantitative and qualitative methods, and were finally calculated using SPSS 22.0 and yaahpV10.3, respectively. Principal component analysis of evaluation indicators was performed using SPSS22.0; a hierarchical analysis diagram was constructed with yahpV10.3 by experts who filled in the judgment matrix of the index layer and the criterion layer, according to which matrix the system assigned the weight of each index and conducted a consistency test, based on which the judgment matrix was adjusted until the optimum results of the consistency test were obtained.

The evaluation score of cultivated land quality was calculated using the integrated index method [21,63] by the following formula:

$$S = \sum_{i=1}^{n} w_i \bullet f_i \tag{7}$$

where $S$ is the cultivated land quality score; $n$, the number of index factors; $w_i$, the weight of each individual index element; and $f_i$, the score of each individual evaluation index (Table 2).

2.3.3. Analysis of the Changes in Cultivated Land Capacity

This paper explores and analyzes, based on quantitative and qualitative analysis and by spatial autocorrelation analysis [64,65], the spatial characteristics and trends of the cultivated land capacity index obtained by multiplying the quantity of cultivated land by the quality score. A study on the degree of change in cultivated land production-capacity was carried out, preceded by a summary of the cultivated land capacity indices of administrative villages, to conclude that cultivated land with concentrated contiguous areas and high quality scores should be protected as a priority [66]. The commonly used spatial autocorrelation analysis tools are Moran's I (global spatial autocorrelation) [35] and the LISA index (local spatial autocorrelation). Moran's I, a measure of the concentration of the regional cultivated land productivity index at the global scale, indicates positive spatial autocorrelation and convergent clustering of study unit attributes when it is greater than 0, negative spatial autocorrelation and discrete distribution when it is smaller than 0 and no spatial autocorrelation and random distribution when it is 0. The LISA index measures the spatial location and extent of the high and low value agglomerations of the cultivated land

capacity index [67]. In this study, Moran's I and the LISA index were calculated separately in ArcGIS. The calculation formulas were:

$$G_i^* = \frac{\sum_{j=1}^{n} w_{ij}x_j - \overline{X}\sum_{j=1}^{n} w_{ij})}{S\sqrt{\frac{\left[n\sum_{j=1}^{n} w_{ij}^2 - \left(\sum_{j=1}^{n} w_{ij}\right)^2\right]}{n-1}}} \tag{8}$$

$$\overline{X} = \frac{\sum_{j=1}^{n} x_j}{n} \tag{9}$$

$$S = \sqrt{\frac{\sum_{j=1}^{n} x_j^2}{n} - \left(\overline{X}\right)^2} \tag{10}$$

where $G_i^*$ is the statistical Z score; $x_j$, the cultivated land capacity index of evaluation unit $j$; $w_{ij}$, the weight between evaluation units $i$ and $j$; $\overline{X}$, the mean value of capacity index score; $n$, the number of evaluation units; and $S$, the standard deviation of index score.

## 3. Results

### 3.1. Changes in the Amount of Cultivated Land

The land-use types in Long'an County from 2010 to 2020 were mainly cultivated land and forest land (Figure 3). The areas of cultivated land and forest land in 2010 were 67,791.23 hectares (29.40%) and 104,805.18 hectares (45.46%), respectively, and the areas of cultivated land and forest land in 2020 were 46,894.24 hectares (20.34%) and 140,325.43 hectares (60.86%), respectively. During the study period, cultivated land decreased by 20,896.99 hectares (9.06%), forest land increased by 35,520.25 hectares (15.41%) most significantly; meanwhile, other lands decreased most significantly by a net 23,030.09 hectares (9.99%). In terms of land use dynamicity, the cultivated land changed gently (3.08%), the largest increase was accounted for by other garden land (141.40%), and the second largest one, by watered land (23.80%); the most significant decrease was seen in other lands (−9.50%), and the second most significant one, in grassland (−5.63%) (Table 3).

**Table 3.** Land-use quantity changes and land-use dynamicity in Long'an County, 2010–2020.

| Land Use Type | Area Change | | | | | | Land-Use Dynamicity (%) |
| --- | --- | --- | --- | --- | --- | --- | --- |
| | 2010 | | 2020 | | Increase or Decrease | | |
| | Area (Hectares) | Proportion (%) | Area (Hectares) | Proportion (%) | Area (Hectares) | Proportion (%) | |
| Paddy land | 12,907.49 | 5.60 | 11,194.31 | 4.86 | −1713.18 | −0.74 | −1.33% |
| Irrigable land | 20.47 | 0.01 | 69.20 | 0.03 | 48.72 | 0.02 | 23.80% |
| Dry land | 54,863.27 | 23.80 | 35,630.73 | 15.45 | −19,232.53 | −8.34 | −3.51% |
| Cultivated land (subtotal) | 67,791.23 | 29.40 | 46,894.24 | 20.34 | −20,896.99 | −9.06 | −3.08% |
| Orchard | 16,654.31 | 7.22 | 19,028.77 | 8.25 | 2374.47 | 1.03 | 1.43% |
| Other garden land | 264.72 | 0.11 | 4009.86 | 1.74 | 3745.14 | 1.62 | 141.48% |
| Orchard (subtotal) | 16,919.03 | 7.34 | 23,038.64 | 9.99 | 6119.61 | 2.65 | 3.62% |
| Forest land | 104,805.18 | 45.46 | 14,0325.43 | 60.86 | 35,520.25 | 15.41 | 3.39% |
| Grass land | 2650.09 | 1.15 | 1157.50 | 0.50 | −1492.59 | −0.65 | −5.63% |
| Urban land | 1392.58 | 0.60 | 1975.56 | 0.86 | 582.98 | 0.25 | 4.19% |
| Rural residential land | 3869.87 | 1.68 | 3325.77 | 1.44 | −544.09 | −0.24 | −1.41% |
| Other construction land | 3256.19 | 1.41 | 5664.84 | 2.46 | 2408.65 | 1.04 | 7.40% |
| Water and water conservancy establishment land | 5638.54 | 2.45 | 6970.81 | 3.02 | 1332.27 | 0.58 | 2.36% |
| Other lands | 24,238.41 | 10.51 | 1208.32 | 0.52 | −23,030.09 | −9.99 | −9.50% |

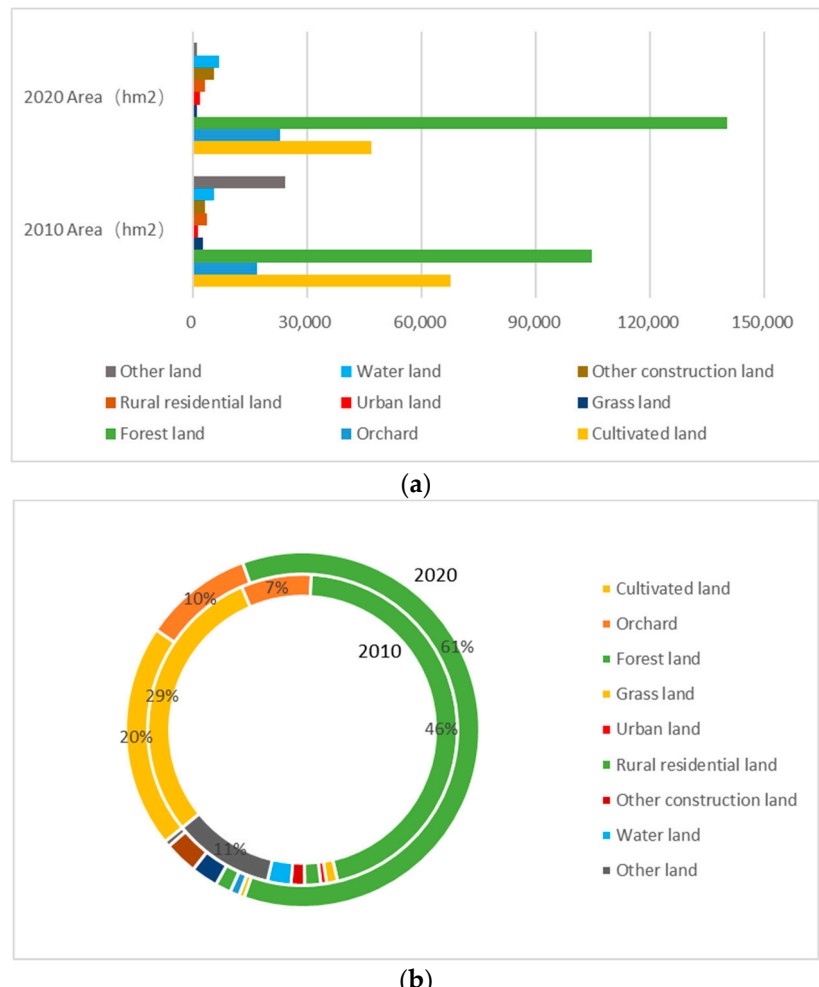

**Figure 3.** (**a**) Comparison of the number of land-use types from 2010 to 2020; (**b**) Comparison of the structure of land-use types from 2010 to 2020.

Figure 3 illustrates the SDE of the cultivated land in Long'an County, from which the concentration and directional trend of cultivated land can be seen. These parameters are reflected by the moving direction of the barycenter and the change in the ellipse area. The cultivated land shows a trend of spatial clustering visualized by the shift of the barycenter's coordinates to the west by 3.3 km (Table 4, Figure 4). While the new cultivated land is mainly concentrated in the south-central part of the county, the reduction in cultivated land is mainly concentrated in the southeast (Figure 5). Based on the calculated average elevation of cultivated land from 2010 to 2020, it could be concluded that the average elevation of cultivated land decreased by 11.91 m from 219.03 m in 2010 to 207.12 m in 2020. The cultivated land patch declined from an average of 2.60 hectares/pc in 2010 to an average of 1.34 hectares/pc in 2020, indicating that the average area of individual patch is decreasing and cultivated land is becoming more fragmented. The regularity of cultivated land patch decreased from 2.08 in 2010 to 1.76 in 2020, indicating an increase in patch regularity, which in turn led to a slight increase in the adaptability of cultivated land to mechanization.

**Table 4.** Variations of SDE parameters.

| Year | $\overline{X}$ | $\overline{Y}$ | $\delta_x$ | $\delta_y$ | $\alpha$ |
|---|---|---|---|---|---|
| 2010 | 36,473,065.06 | 2,556,113.04 | 24,960.18 | 13,206.77 | 97.50 |
| 2020 | 36,469,780.12 | 2,556,057.73 | 24,475.51 | 13,150.12 | 99.36 |
| Difference | −3284.94 | −55.31 | −484.67 | −56.65 | 1.85 |

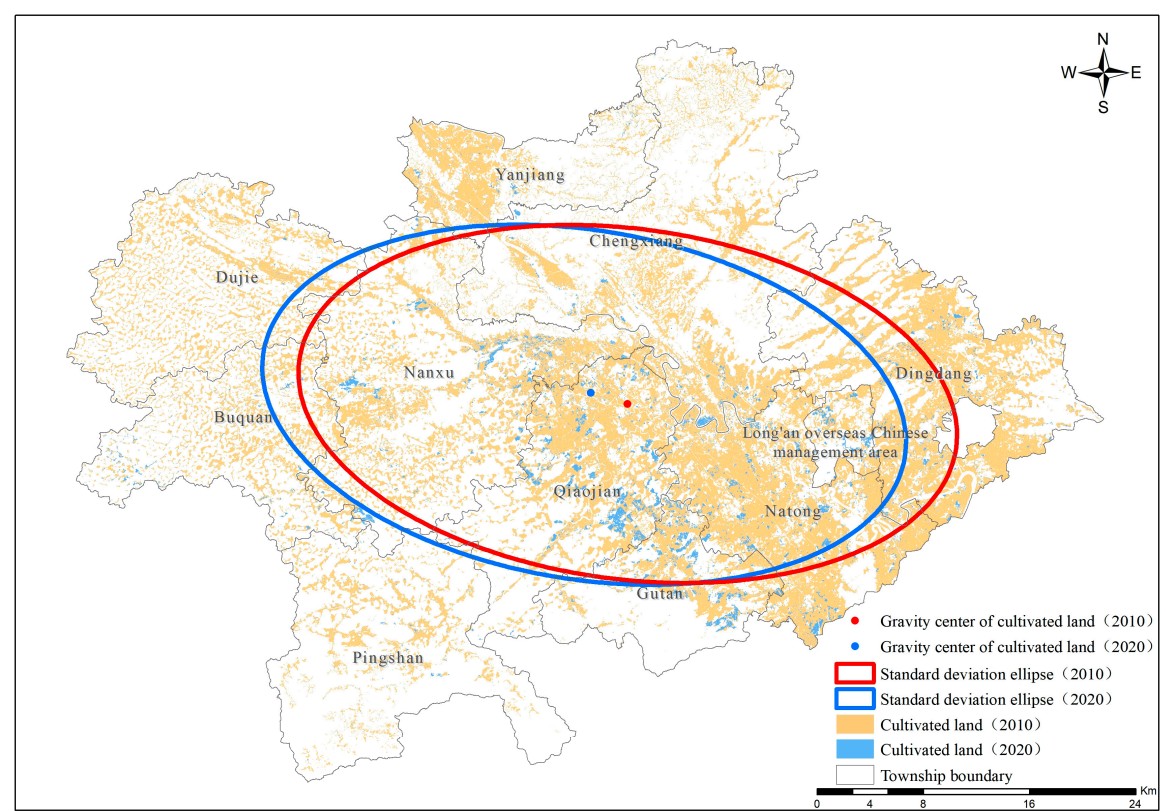

**Figure 4.** SDEs of cultivated land (2010–2020).

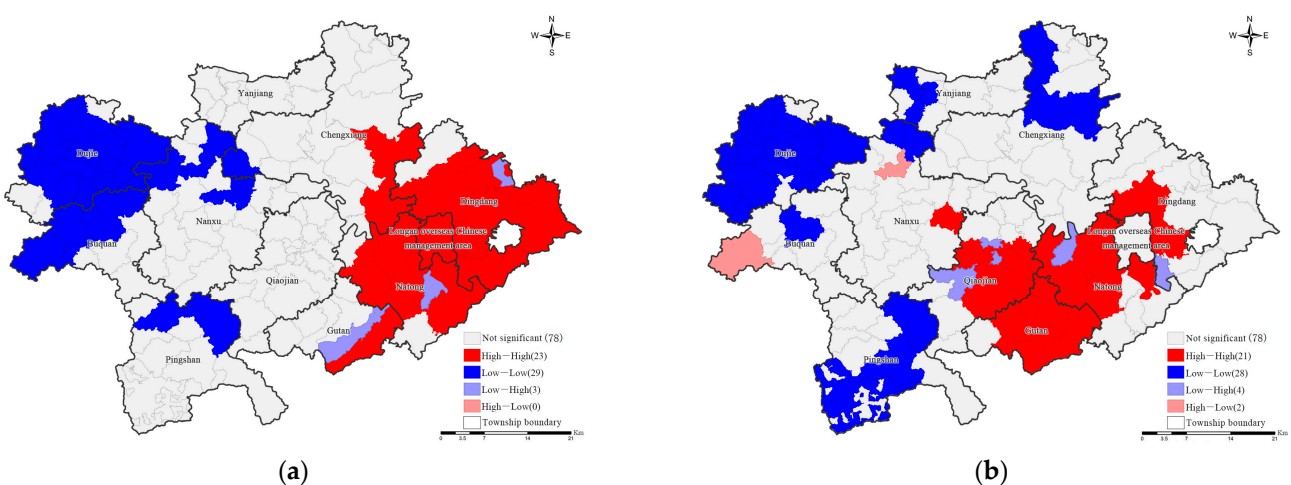

(**a**)        (**b**)

**Figure 5.** (**a**) Cluster distribution of cultivated land reduction from 2010 to 2020; (**b**) Cluster distribution of cultivated land addition from 2010 to 2020.

In this table, $\overline{X}$ and $\overline{Y}$ are the horizontal and vertical coordinates of the standard deviation of the ellipse, respectively; $\delta_x$ and $\delta_y$ are the standard deviations along the x and y axes, respectively; $\alpha$ is the orientation angle.

In order to explore the structural changes in cultivated land, the specific flows be-
tween cultivated land and other land types were analyzed using a land conversion matrix
(Figure 6). From 2010 to 2020, the cultivated land changed so significantly that its total
area of 67,791.23 hectares in 2010 changed, by a net outflow of 20,896.99 hectares between
an outflow (decrease) of 27,341.07 hectares and an inflow (increase) of 6444.08 hectares
during the ten-year period, into the total area of 46,894.24 hectares in 2020. Among the
total outflow, the dry land, water land and paddy land account for 24,773.81 hectares
(90.61%), 9.91 hectares (0.04%) and 2557.36 hectares (9.35%), respectively. The dry land
mainly became orchards, forest land and other garden land, accounting for 44.75%, 34.74%
and 5.16% of the total area of lost dry land, respectively. Irrigable land mainly became dry
land, forest land and other construction land, accounting for 45.28%, 15.49% and 12.63%,
respectively. Paddy land mainly became forest land, orchard land and water and water
conservancy facilities land, accounting for 24.48%, 22.73% and 18.60%, respectively. During
the study period, other lands, such as dry land (5541.28 hectares, 85.99%), watered land
(58.63 hectares, 0.91%) and paddy land (844.17 hectares, 13.10%) changed to cultivated
land. The land types changed into dry land were mainly orchard land, forest land and
paddy land, accounting for 56.77%, 15.76% and 7.31%, respectively; the land types changed
into watered land were mainly dry land, paddy land and orchards, accounting for 45.43%,
30.25% and 20.96%, respectively; the land types changed into paddy land were mainly dry
land, water and water conservancy facilities land and forest land, accounting for 70.43%,
8.74% and 7.07%, respectively.

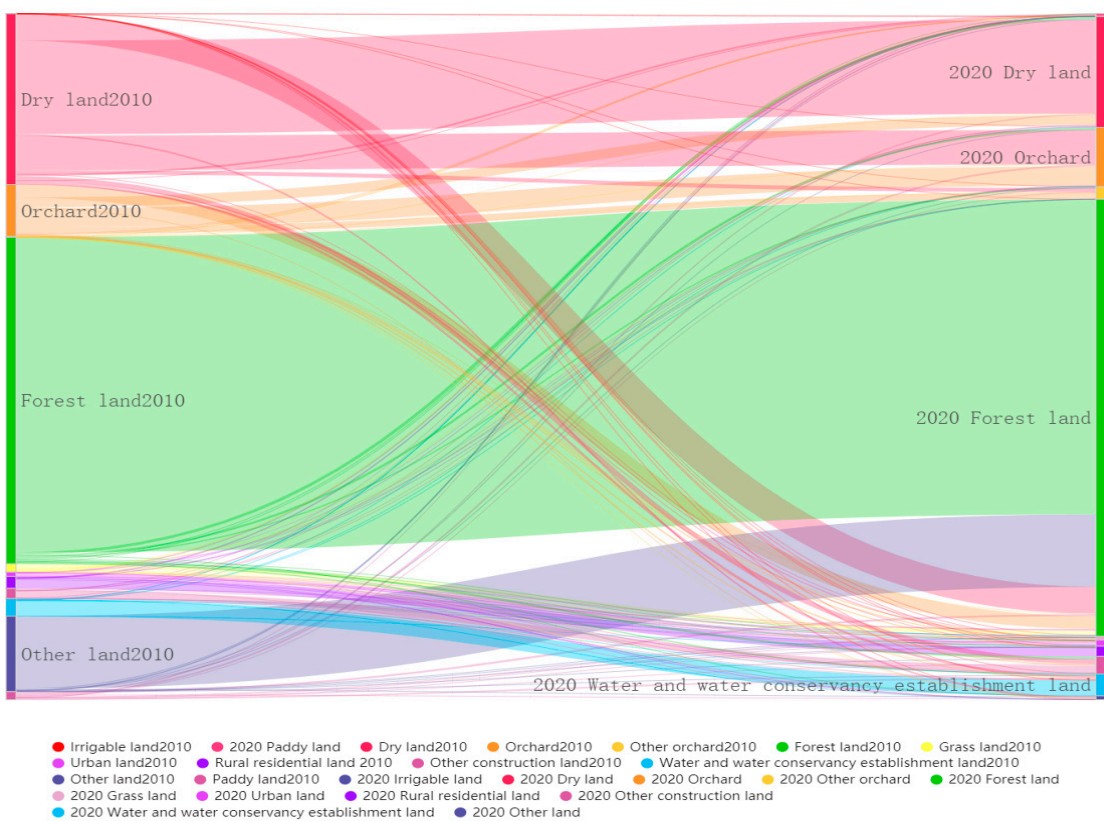

**Figure 6.** Inflow and outflow of all land types from 2010 to 2020.

### 3.2. Changes in Cultivated Land Quality

According to the evaluation index scores and their weights (Table 2), a weighted
superposition analysis was conducted and a cultivated land quality score was calculated
by Equation (7). The higher the quality score of the cultivated land in the evaluation unit is,
the better the quality of the cultivated land is. The results showed that the quality scores
of cultivated land in the study area were between 57.27 and 91.81. The cultivated land in

the study area was classified into six grades using the natural breakpoint method (Table 5) based on the composite score: ≥82 for Grade 1, ≥77~82 for Grade 2, ≥73~77 for Grade 3, ≥69~73 for Grade 4, ≥65~69 for Grade 5 and ≤65 for Grade 6. Figure 7a,b illustrates the distribution of quality grades of cultivated land in the study area from 2010 to 2020.

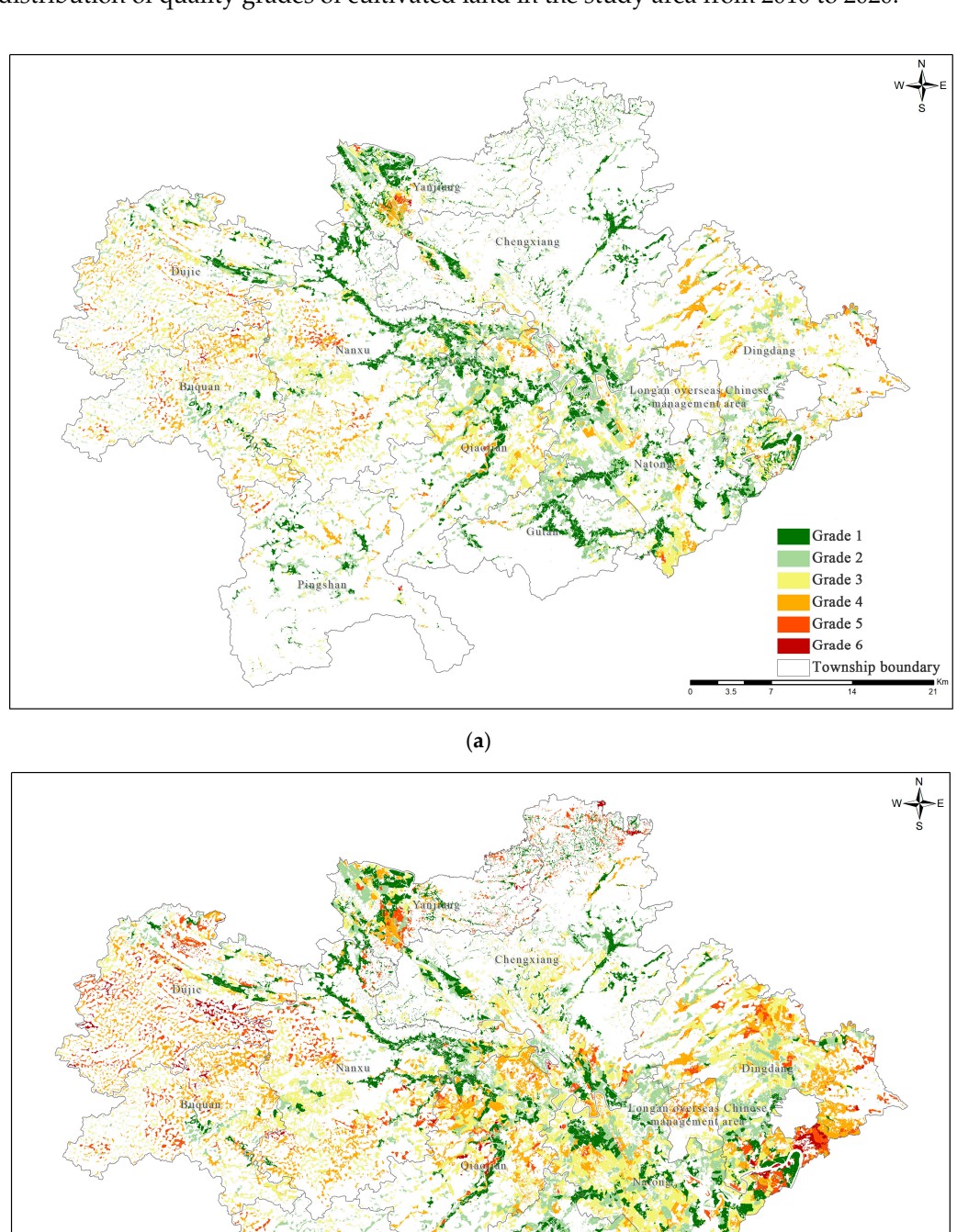

(**a**)

(**b**)

**Figure 7.** (**a**) Quality grading map of cultivated land 2010; (**b**) Quality grading map of cultivated land 2020.

**Table 5.** Area and proportion of cultivated land quality of different quality grades, 2010–2020.

| Cultivated Land Grade | Cultivated Land Area and Its Proportion | | | | | |
|---|---|---|---|---|---|---|
| | 2010 | | 2020 | | 2020–2010 | |
| | Area (Hectares) | Proportion (%) | Area (Hectares) | Proportion (%) | Area (Hectares) | Proportion (%) |
| Grade 1 | 11,879.80 | 17.52% | 11,301.69 | 24.10% | −578.11 | 2.77% |
| Grade 2 | 13,993.28 | 20.64% | 13,590.35 | 28.98% | −402.93 | 1.93% |
| Grade 3 | 20,428.58 | 30.13% | 14,120.60 | 30.11% | −6307.98 | 30.19% |
| Grade 4 | 14,545.08 | 21.46% | 6589.68 | 14.05% | −7955.41 | 38.07% |
| Grade 5 | 5825.54 | 8.59% | 1244.14 | 2.65% | −4581.40 | 21.92% |
| Grade 6 | 1118.95 | 1.65% | 47.78 | 0.10% | −1071.17 | 5.13% |
| Total | 67,791.23 | 100.00% | 46,894.24 | 100.00% | −20,896.99 | 100.00% |

As indicated by the results, the quality distribution of cultivated land in the study area shows a general trend that the quality of the central-eastern valley and the plain of peaks and monadnocks is better than the quality of the surrounding areas, and that the high-quality cultivated land is concentrated in the central zone along the course from northwest to southeast, while the quality of cultivated land in the surrounding areas is inferior to that in the central area. In 2010, the proportions of cultivated land of Grades 1, 2, 3, 4, 5 and 6 in the total cultivated land in the study area were 17.52%, 20.64%, 30.13%, 21.46%, 8.59% and 1.65%, respectively. In 2020, such proportions were 24.10%, 28.98%, 30.11%, 14.05%, 2.65% and 0.10%, respectively (Table 5). The quality of cultivated land in Long'an County displays an upward trend during the 10-year period. In 2020, the largest area of cultivated land was Grade 3, which indicates that most of the cultivated land is above medium grade. During the study period, the overall scores of the evaluation indicators in 2020 increased by 2.71% compared with those in 2010, and the average scores of the evaluation factors of cultivated land show that the average scores of the natural, socio-economic, and ecological factors of cultivated land increased by 4.07% most significantly, 0.49% and 1.56%, respectively. The increase in the average scores of both soil PH and organic matter content by 11.87% and 10.30%, respectively, compared with those in 2010 was the main reason for the score increase for natural factors. The increase in the average score of ecological land coverage by 25.37% compared with 2010 was the main reason for the score increase for ecological factors. The decrease in the average score of patch size by 22.32% compared with that in 2010 was the main reason for the score decrease for socio-economic factors (Table 6).

The net decrease in high-quality cultivated land (Grade 1 and Grade 2) in the study area is 981.04 hectares, accounting for 4.69% of the total reduction area. Though scattered, the net decrease is relatively concentrated around the Nanxing reservoir in Nanxing Village, Nanxu Town, Nafa Village, Chengxiang Town, Yuying Village and Macun Village, Gutan Township, Xiadeng Village, Natong Town, Baowan Village and Dingdang Town and mostly it is planting structure adjustment. The net decrease (14,263.39 hectares) in medium grade land (Grade 3 and Grade 4), accounting for 68.26% of the total decreased area, is relatively concentrated, mainly in the northeastern of Qiaojian Town, Natong Town, the northern area of the Overseas Chinese Administrative Zone, the central–eastern area of Dingdang Town and mostly it is also planting structure adjustment. The net decrease (5652.56 hectares) in poor cultivated land of Grade 5 and Grade 6, accounting for 27.05% of the total decreased area, is relatively concentrated in the southeast of Dingdang town. The new cultivated land, which is little, mainly consists of Grade 2 and Grade 3 and it is relatively scattered in the center of the county and relatively concentrated in the eastern of Nanxu Town, the north of Qiaojian Town and Gutan Township, and Natong Town.

**Table 6.** Comparison of average scores of evaluation indicators for cultivated land, 2010–2020.

| Evaluation Objectives | Evaluation Factors | Evaluation Indicators | 2010 | 2020 | Percentage of Change |
|---|---|---|---|---|---|
| Cultivated land Quality | Natural Factors | Average slope | 14.94 | 15.21 | 1.81% |
| | | Effective soil layer thickness | 6.93 | 6.93 | 0.00% |
| | | Surface soil texture | 8.13 | 8.19 | 0.74% |
| | | Organic matter content | 9.03 | 9.96 | 10.30% |
| | | Soil PH | 3.96 | 4.43 | 11.87% |
| | | Subtotal | 42.97 | 44.72 | 4.07% |
| | Socio-economic Factors | Irrigation guarantee rate | 5.14 | 5.41 | 5.25% |
| | | Drainage conditions | 4.29 | 4.29 | 0.00% |
| | | Density of field roads | 1.03 | 1.03 | 0.00% |
| | | Cultivation distance | 5.21 | 5.11 | −1.92% |
| | | Patch size | 1.12 | 0.87 | −22.32% |
| | | Patch regularity | 1.47 | 1.64 | 11.56% |
| | | Subtotal | 18.33 | 17.55 | 0.49% |
| | Ecological Factors | Ecological land coverage | 2.05 | 2.57 | 25.37% |
| | | Soil erodibility K value | 6.09 | 6.09 | 0.00% |
| | | Soil pollution level | 6.61 | 6.31 | −4.54% |
| | | Subtotal | 14.74 | 14.97 | 1.56% |
| | Total | | 75.98 | 78.04 | 2.71% |

### 3.3. Change in Production Capacity of Cultivated Land

In this paper, the Moran scatter plots displaying the Moran's I values calculated for the production capacity indices of cultivated land on a village-by-village basis are shown in Figure 8. The Moran's I indices for 2010 and 2020 were 0.5466 and 0.3753, respectively, both of which passed the significance test. These results confirm the positive spatial correlation of the capacity index of cultivated land, namely, neighborhood is a factor in the coordination of capacity. Hotspot analysis was carried out to identify where elements with high and low values of overall capacity of cultivated land cluster spatially [53,59]. Based on another hotspot, an analysis was performed using the LISA index [68], the graphs of spatial autocorrelation types were generated to represent the locally spatial correlation of Moran's I values [69,70]. Each graph was then divided into four quadrants: quadrant 1 (high, high), quadrant 2 (low, high), quadrant 3 (low, low) and quadrant 4 (high, low) where high values are surrounded by high values in quadrant 1, low values are surrounded by high values in quadrant 2, low values are surrounded by low values in quadrant 3 and high values are surrounded by low values in quadrant 4. The (high, high) and (low, low) quadrants indicate local spatial positive correlation within the local space, while the (low, high) and (high, low) quadrants indicate local negative correlation within local space.

After the hotspot analysis approach was used to locally cluster the capacity indices of cultivated land, the results were divided into seven grades where the best capacity was found within the high-high and high-medium adjacent clusters, which were classified into key protection; the better capacity was found within the high-low, medium-medium and non-significant clusters, which were classified into secondary protection; and the worst capacity was found within the low-low and low-medium adjacent clusters, which were classified into tertiary protection. The results of locally spatial autocorrelation are displayed in Figure 9 where the high-high and low-low types indicate not only positive spatial correlations between the index of village production capacity and the type of neighborhood but also consistent evolutionary trends, the low-high and high-low types indicate that the scores of village-level production capacity are spatially negatively correlated with the indices of production capacity of surrounding villages, namely, the production capacity index of the target village is opposite to those of the surrounding villages. The high-low and low-high types are spatially similar to "high land" and "low land". According to the theory of spatial polarization, the target area will eventually be assimilated by the surrounding areas and then transformed into a high-high or low-low type. Therefore, spatial correlation is worthy of attention in the conservation and sustainable use of cultivated land.

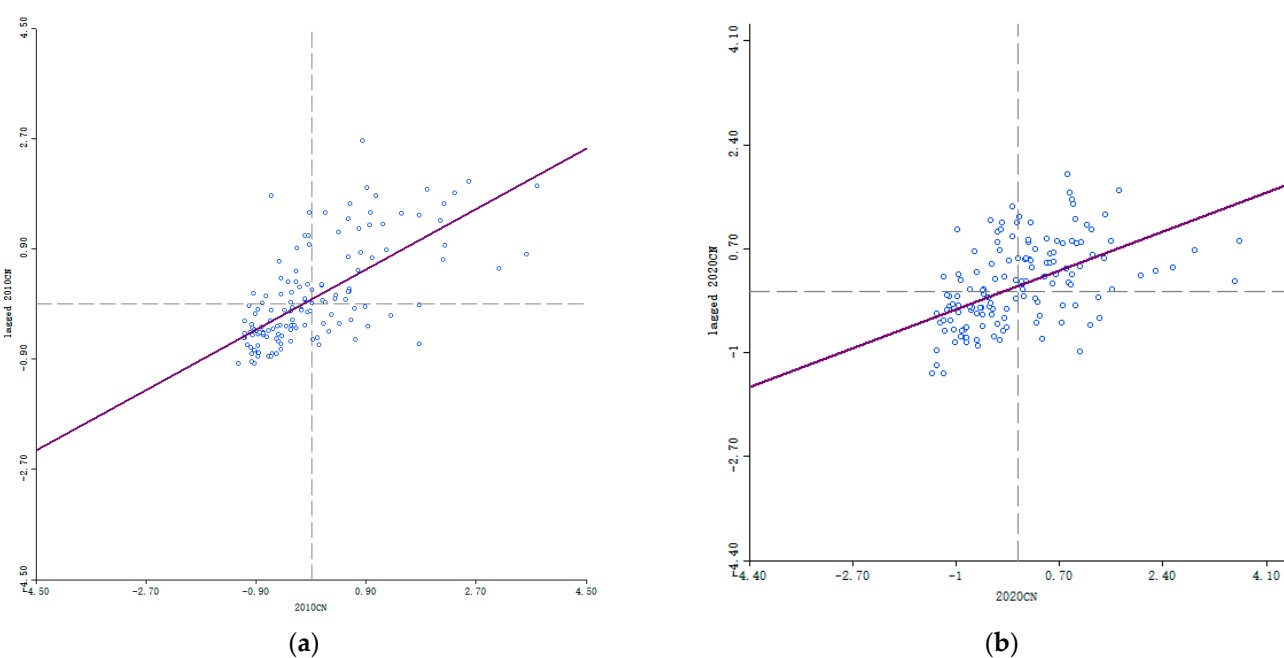

**Figure 8.** (**a**) Scatterplot of global Moran indices for capacity scores in 2010; (**b**) Scatterplot of global Moran indices for capacity scores in 2020.

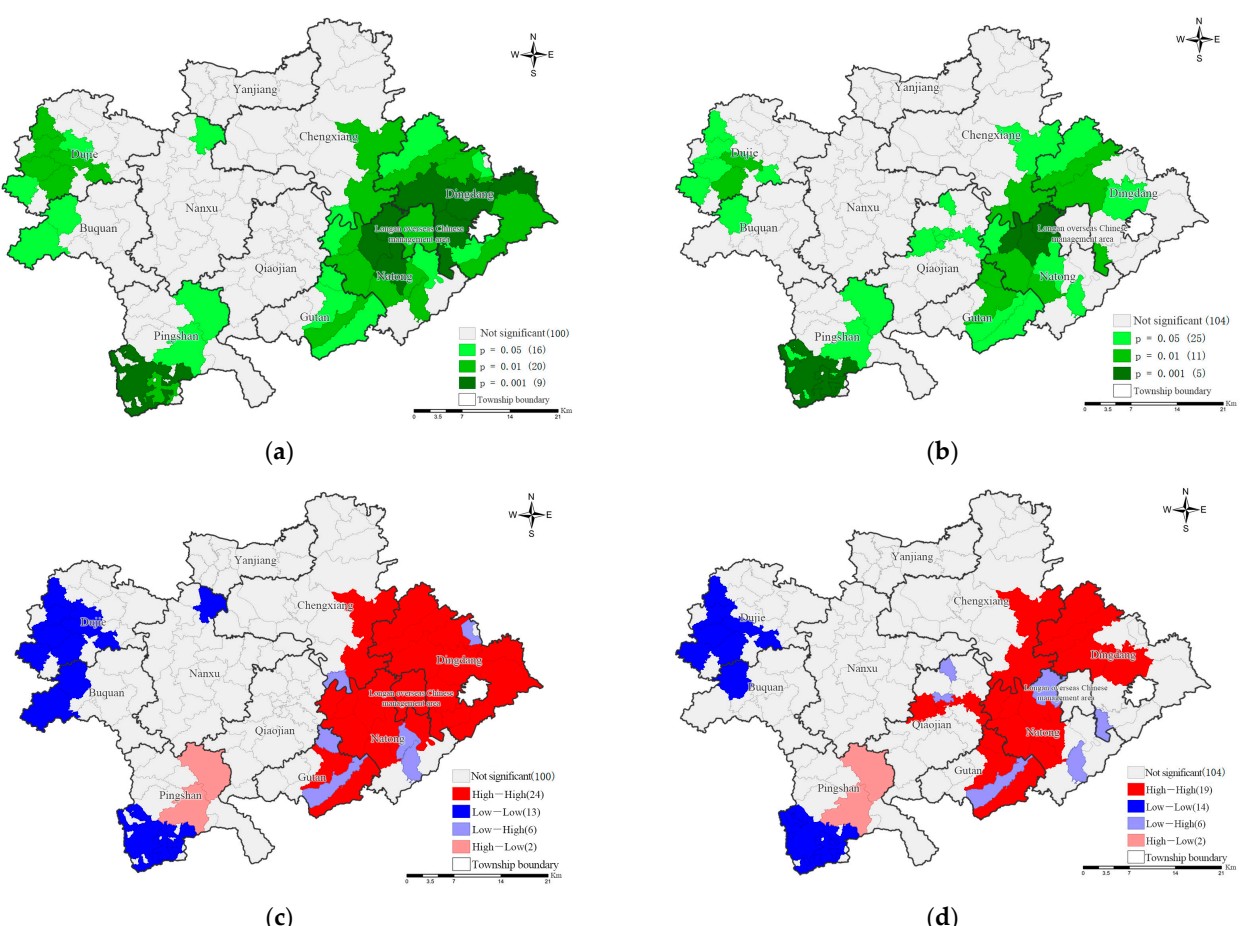

**Figure 9.** (**a**). Significance grading chart for 2010; (**b**). Significance grading chart for 2020; (**c**). Aggregation based on locally spatial correlation for 2010; (**d**). Aggregation based on locally spatial correlation for 2020.

In addition, Figure 9 shows that the spatial clustering based on both annual production capacity scores is not obvious in most areas of the Chengxiang Town, Yanjiang Town, Nanxu Town and Qiaojian Town. Most of the high-value areas are concentrated in Dingdang Town, Natong Town and parts of the Overseas Chinese Administrative Zone, while the low-value areas are distributed in the remote western areas (Dujie Township, Buquan Township and Pingshan Township). The low-high and high-low towns, which are spatially negatively correlated, are sporadically distributed.

## 4. Discussion

This paper explores the spatio-temporal patterns of cultivated land changes at a small county scale by analyzing the quantity, quality, production capacity and ecology of cultivated land from 2010 to 2020. The results show that the overall conversion of cultivated land into other land types in the study area from 2010 to 2020 is of great significance, and the main reason for the cultivated land decrease is not urban expansion but the outflow from cultivated land into other agricultural lands. The ten-year period has seen the quality of cultivated land slightly improve, the production capacity of cultivated land show a downward trend, the barycenter of cultivated land shift to ecologically sensitive areas, the patch fragmentation of cultivated land has aggregated, the patch regularity has improved and the adaptability of cultivated land to mechanization has slightly improved.

### 4.1. Changes in the Amount of Cultivated Land

During the study period, cultivated land significantly decreased by 27,341.07 hectares, and the lost cultivated land mostly became orchard land, forest land and other garden lands. The main reasons therefore include: firstly, Long'an County lies to the south of the Tropic of Cancer and under a hot and rainy climate that is very suitable for the cultivation of subtropical fruits, and Long'an County has the tradition of planting bananas, oranges and other local cash crops. Secondly, cultivated land is left unattended after the farmers go out to work as migrant workers. Thirdly, some areas, due to high altitude and irregular shape, are not convenient for cultivation, prompting the farmers to quit from cultivation or convert cultivated land to forest land because of the small margin for food planting and the obvious marginalization of cultivated land. The total new cultivated land is 6444.08 hectares, mainly from orchard land and forest land, and few from grassland, construction land and other lands. The original garden land partially restored into cultivated land constitutes the main source and accounts for 57.31% of the new cultivated land. Some residual and inferior forest lands, abandoned residential and mining lands, grasslands, pits and ponds have been consolidated into new cultivated land, accounting for 30.99% of the new cultivated land. Despite the new cultivated land accounting for 23.57% of the decrease in cultivated land, there is a net decrease of 20,869.66 hectares of cultivated land. The net decrease in urban construction land, rural residential land and other construction land is 1775.27 hectares, which is much less than the area of new cultivated land in the same period. This shows that the "balance of cultivated land between occupation and replenishment" implemented in Long'an County has worked if not considering the outflow from cultivated land into other agricultural lands.

### 4.2. Changes in Cultivated Land Quality

During the study period, the overall quality of cultivated land in Long'an County showed an upward trend, and most pieces of cultivated land were middle and high grades. During the 10-year period, the authorities of natural resources of both Long'an County and Guangxi Province invested a lot of money and implemented several batches of land consolidation projects. Through land consolidation, the infrastructure and quality of cultivated land have been improved. Improvement of soil PH value and increase in organic matter content are the main indicators of the better quality of cultivated land. Smaller patch size that aggravated soil pollution and a longer cultivation distance are the main reasons for the decrease in cultivated land quality.

*4.3. Changes in Cultivated Land Production Capacity*

Through the spatial autocorrelation analysis, it can be observed that it is Dingdang Town, Natong Town and the Overseas Chinese Administrative Zone where the lands with high production capacity are concentrated, thus these areas should be better protected to ensure that the capacity of cultivated land does not decrease. The lands with a low value of production capacity are concentrated in the remote areas of Dujie Township, Buquan Township and Pingshan Township, where inferior cultivated lands are relatively concentrated and need increased capital investment to improve production capacity. The fragmentation of cultivated land in the study area is exacerbated because the lost cultivated lands are mostly concentrated and contiguous, while the new cultivated lands are more scattered. Most new cultivated lands are in ecologically sensitive areas and play an important part in ecosystem services, as a result of which it is likely that the value generated by crops on the new cultivated lands will not be able to compensate for the value of lost ecological services. From the ecological environmental perspective, this is unsustainable and does not allow for an optimistic view on the ecological impact.

Based on the above results, we suggest that authorities of natural resource management should further regulate the conversion of cultivated land into other agricultural lands and strengthen the supervision of the internal structural transformation of cultivated land. The implementation of the policy, i.e., the balance of cultivated land between occupation and replenishment, has effectively restrained the conversion from cultivated land into construction land and basically curbed the "non-agriculturalization" of cultivated land. During the study period, most pieces of lost cultivated land were changed into orchard land and forest land, and although food production capacity decreased, the cultivated layer of the cultivated land was not destroyed. Business diversification by the farmers has helped to improve their income and contributed to rural revitalization in underdeveloped areas. It is recommended that differentiated policies on cultivated land protection should be formulated and implemented, and that while national food security is emphasized, farmers' income and livelihoods should also be given due consideration as to appropriate planting restructuring of cultivated land. The analysis shows that the overall quality of cultivated land has increased during the study period because of artificial empowerment and agricultural restructuring. It is recommended to continue to encourage the development of high-standard farmland, and comprehensive consolidation of farmland in the whole area or other projects to achieve greater area, quality and the ecological grade of cultivated land.

Some limitations can be improved in the future. Firstly, in the evaluation of cultivated land quality, the selection of indicators is not comprehensive enough due to the limited availability of data, and the selection of indicators for the ecology and soil pollution of cultivated land needs to be improved. Secondly, in regard to the production capacity of cultivated land, the hot spot analysis, which was based on the production capacity indices obtained by multiplying cultivated land quantity with the comprehensive score of cultivated land, was not intuitive enough without any correspondence established between cultivated land gradation and specific grain yield. In addition, an in-depth analysis of the changes in all land types in the study area is expected to discover the change patterns among these types and explore the evolutionary mechanism behind them, identify the optimal combination of increase in farmers' income and protection of cultivated land and provide a guarantee of resources for the sustainable development of the county.

## 5. Conclusions

In the context of rapid urbanization and agricultural restructuring, the contradiction among economic development, agricultural restructuring and cultivated land protection has intensified and food security is faced with challenges. Therefore, while ensuring economic development and farmers' income increase, we have to endeavor towards the common of goal food security. The spatial and temporal evolution patterns of cultivated land in Long'an County were studied based on the ten-year National Land Survey data and cultivated land quality data. The results of the study are as follows: during 2010–2020,

the total amount of cultivated land in Long'an County underwent a sharp decrease that accounts for 30.83% of the total cultivated land area in 2010. The lost cultivated land mainly has changed into orchard land, forest land and other garden land; consequently, the main reason for the decrease in cultivated land is not the expansion of urban land, but the conversion of cultivated land into other agricultural lands. The overall quality of cultivated land in the study area was improved by 2.71%. Average scores of the natural, socio-economic and ecological factors were increased by 4.07% most significantly, 0.49% and 1.56%, respectively. The increase in cultivated land quality is mainly due to artificial empowerment, which results in better organic matter content, soil PH, ecological land coverage and patch regularity; the fragmented cultivated land (which was caused by the decrease in patch size), aggravated soil pollution and longer cultivation distance are the main reasons for the reduction in cultivated land quality. The overall food production capacity showed a decreasing trend that includes a drop of 28.96%. Despite the better quality of cultivated land, it is not enough to compensate for the loss of food production capacity because of the significant reduction in the quantity of cultivated land. Along with the 3.3 km-long movement of the barycenter of cultivated land to the west (ecological zone), there is a decrease in the average score, in the case of patch size of cultivated land, by 22.32%, soil pollution 4.54% and cultivate distance 1.92%. While sustained stability of cultivated land decreased, the patch fragmentation of cultivated land showed an upward trend. There are two reasons for the lower total food-production capacity in Long'an County: first, the low comparative income of grain cultivation, because of which farmers spontaneously adjusted the agricultural cultivation structure to pursue high returns; and second, the lack of targeted government policies to protect cultivated land. In general, this comprehensive analysis method can be applied to other provinces in China or other regions abroad to provide a reference basis for a comprehensive understanding of changes in food production capacity of cultivated land and the formulation of policies on cultivated land protection.

**Author Contributions:** Supervision, K.W. and W.Y.; project administration and funding acquisition, S.L.; conceptualization and writing—original draft preparation, J.D.; writing—review and editing, J.D. and B.L.; software and data curation, Q.L.; conceptualization, methodology, visualization, and supervision, S.L. and Q.L.; All authors have read and agreed to the published version of the manuscript.

**Funding:** This research was funded by National Key Research and Development Program (Grant No. 2021YFD1500203) and the Key Laboratory of Land Satellite Remote Sensing Application, Ministry of Natural Resources of the People's Republic of China (Grant No. KLSMNR-202205).

**Institutional Review Board Statement:** Not applicable.

**Informed Consent Statement:** Not applicable.

**Data Availability Statement:** The datasets used and/or analyzed during the current study are available from the corresponding author upon reasonable request.

**Conflicts of Interest:** The authors declare no conflict of interest.

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
