# Peer review of "Spatio-Temporal Analysis of Cultivated Land from 2010 to 2020 in Long’an County, Karst Region, China"

_land, doi:10.3390/land12020515_

Round 1
Reviewer 1 Report
The paper titled “Comprehensive Analysis of Spatial and Temporal Changes in cultivated land: A Case Study of Longan County, Guangxi Province, China” strived to construct a comprehensive analytical framework to study the changes in cultivated land of a county in ten years using the trinity theory and spatial analysis using GIS methods. The results found that the total amount of cultivated land in the study decreased sharply and the decreased cultivated land mainly changed to orchards and forests. The quality of cultivated land increased, and the total food production capacity of cultivated land decreased, mainly due to the area decrease of cultivated land. Both the ecological grade and the sustainability of cultivated land decreased slightly. The barycenter of cultivated land migrated 3.3 km to the ecologically sensitive areas in the west, and the patch size of cultivated land decreased, with increased fragmentation of cultivated land.
The innovation of this paper is weak, the authors sure know that published works of literature have investigated the cultivated land changes using the trinity method and framework. The authors stated that the study can provide a reference for “other countries” to understand the changes in cultivated land. This statement is not appropriate. Even though the authors used more precise land use data, the DEM data is not high-resolution, which impacts the convincing and precision of the results. The authors may want to modify the expression to avoid boasting.
Besides, the quality of English writing and expression in this paper is below the standard of land. Help from native professional speakers is recommended.
Other comments:
1. The paper has many mistakes, e.g. Line 8 and 9, the starting serial numbers should be 3 and 4.
2. Line 10, there are two “Correspondence:”
3. Line 18, “different periods”? As I found in the paper, the study period is from 2010 to 2020.
4. Line 23, “The results suggest that”, the authors may want to revise as “The results demonstrated that”.
5. Introduction, the authors may want to rewrite to thoroughly cover the current study trends of the trinity method.
6. Line 98, “karst and poor regions” can be modified as “karst and underdeveloped regions”.
7. Line 120, 134, and 624, “other countries” is not appropriate.
8. Line 136, Figure 1, “different periods”?
9. From Figure 1, the authors separately estimated the quantity, quality, and ecology of cultivated land, how do the authors show the “comprehensive analysis”?
10. Line 164, Figure 2, the authors used “Long’an” to name the study area, while in the text it is “Longan”.
11. Figures 2 to 9, the Legends are too small, the authors may want to modify them to make them clear.
12. Line 268, Table 2, how did the authors assign the score of each indicator?
13. Line 507, “in two periods”.
14. The authors did not list the names of the second and third authors in most of the references, and there are small mistakes in the Reference section (e.g. 51).
Reviewer 2 Report
The article “Comprehensive Analysis of Spatial and Temporal Changes in cultivated land: A Case Study of Longan County, Guangxi Province, China” presents relevant material regarding changes in cultivated land during a ten-year period, paying special attention to production capacity and ecology.
Observations:
1. The article presents the material from 2010-2020 research in Guangxi province, China. There should be a clearer justification why this particular province was selected for the research, and whether the results can be applied to other provinces in China, or even other countries.
2. I suggest to pay greater attention to the quality of pictures, information presented is difficult to understand at times (Figure 2. b and c, Figure 4. a and b, Figure 7. a, b, c, d, e, f, g, h, i, j, k and particular).
Reviewer 3 Report
The manuscript analyses the characteristics of spatial and temporal changes in cultivated land in Longan province, Guangxi County, China, using different indexes, measures, and visualizations. This is an interesting study and a complete work with an extensive literature review and comprehensive analyses. The topic fits the scope of the journal and the case is relevant. The manuscript describes valuable applied research which has practical value, the results and methods used are clearly presented, references are current, and the experimental design is appropriate. This work is an excellent contribution to the existing knowledge.
To my opinion the article is interesting and I propose to accept it in its present form. However, I addressed a few remarks and propose to make some minor tweaks before publishing:
Figure 1: explain or rephrase the name of the data source: “quality level update data”
Figure 2, b & c – improve resolution and resize legend and other text; it is unreadable
All Tables: align and format numbers correctly – column align decimal symbols to easier compare the results.
I suggest using hectares or square kilometers instead of hm.
Figures 5 & 7: improve resolution, smaller images are completely unreadable. Is it necessary to include all those images?!
Reviewer 4 Report
The manuscript entitled "Comprehensive Analysis of Spatial and Temporal Changes in cultivated land: A Case Study of Longan County, Guangxi Province, China " corresponds to the journal's subject areas.
The scientific idea of the manuscript and its presentation are quite well presented. It is quite clear what the authors wanted to say in their study. However, the quality of the illustrations is quite poor. Several revisions should be made before the manuscript can be accepted for publication. Below you can find my main comments:
The aim of the work is not given in the abstract.
Figure 1 is not referenced in the text.
The font in Figure 2 is too small. The reader cannot see anything.
Who is the author of the formulas presented in the methodology section.
Explanations of what those symbols mean should be provided under Table 4. This would make it easier for the reader, as there would be no need to go back to the methodology section.
The font in Figure 5 is very small. It is difficult to see well.
Lines 390-392: Based on what criteria was chosen to use the assessment from Grade 1 (>82) to Grade 6 (<65). Why 6 Grades?
The font in Figure 7 is very small. It is difficult to see well.
The font in Figure 9 is very small. It is difficult to see well.
The discussion section could have similar subsections as the results section. It would be clearer for the reader.
Round 2
Reviewer 1 Report
I am happy to see that all the comments were correctly addressed by the authors.